# Taming Large Language Models for Free-Form Generation Via Reinforcement Learning With Verifiable Rewards

## Abstract

Evaluating open-ended free-form generation is challenging because it is hard to define what clearly separates good from bad outputs. Existing methods often miss key aspects like coherence, style, or relevance, or are biased by pretraining data, making open-ended long-form evaluation an underexplored problem. To address this gap, we propose semantic evaluation, a scoring model using an LLM as reward model for evaluating open-ended free-form generation in GRPO and guiding its training to produce enough distinct rewards for good and bad outputs. Through comprehensive evaluations, including LLM-as-a-judge, human ratings, and qualitative analysis, we show that using LLM scorers trained on multi-sentence and paragraph-length responses, remains more reliable across varied long passages and aligns well with the verifiable rewards GRPO needs than standard free-form metrics. Human evaluations confirm that using trained LLM rewards as the reward signal to train policy models yields responses better aligned with human preferences than those trained with traditional metrics.

## 1 Introduction

Identifying the good and bad generations is the key to the success of reinforcement learning with verifiable rewards (RLVR) to improve LLMs' abilities on structured rule-based tasks such as mathematical problem-solving (Ahn et al., 2024), classification (Rouzegar & Makrehchi, 2024), and game planning (Shao et al., 2019). RLVR algorithms like Group Relative Policy Optimization (GRPO) excel in mathematical problem-solving tasks by leveraging clear, rule-based reward signals (e.g., correctness) that effectively distinguish between correct and incorrect responses (DeepSeek-AI et al., 2025). However, despite these advances, extending GRPO to open-ended, free-form text generation tasks, such as generating travel plans, creative compositions, instruction-following, remains largely underexplored. A key challenge in applying GRPO to open-ended generation is the lack of quantifiable evaluation criteria, making it difficult to distinguish high-quality responses from poor ones. (Krishna et al., 2021). Unlike structured tasks evaluated by explicit correctness criteria, free-form generation requires models to balance coherence, fluency, and informativeness which are all subjective dimensions that are hard to define with discrete rules. This makes reward design a fundamental challenge in GRPO: How do we guide a model to write better, not just longer?

A key barrier is the lack of reliable, scalable evaluation methods. Popular metrics like ROUGE (Lin, 2004) and BERTScore (Zhang et al., 2020) that measure lexical or embedding-level similarity to a reference often fail to capture qualities that humans actually care about, such as clarity, relevance, and stylistic appropriateness (Chen et al., 2019; Li et al., 2024). Thus, traditional metrics correlate poorly with human preferences, making them unreliable reward signals in GRPO pipelines. Recent work has applied general reward models (GRM) (Lambert et al., 2024) in RLHF to select preferred responses, but they are not designed as verifiable rewards for GRPO, and their large size introduces latency and high computational cost. As shown in Figure 1, traditional metrics fail to distinguish between clearly better and worse responses, assigning similar scores to both, while GRM rewards often correlate with length rather than content quality.

To address open-ended evaluation in GRPO, we propose FreeForm-RM, a LLaMA 3-based (Grattafiori et al., 2024) model trained on diverse free-form responses and human ratings to provide free-form response quality evaluation. We hypothesize that trained FreeForm-RM can provide stronger reward signals than traditional metrics, enabling GRPO to guide models that generate more meaningful free-form responses.

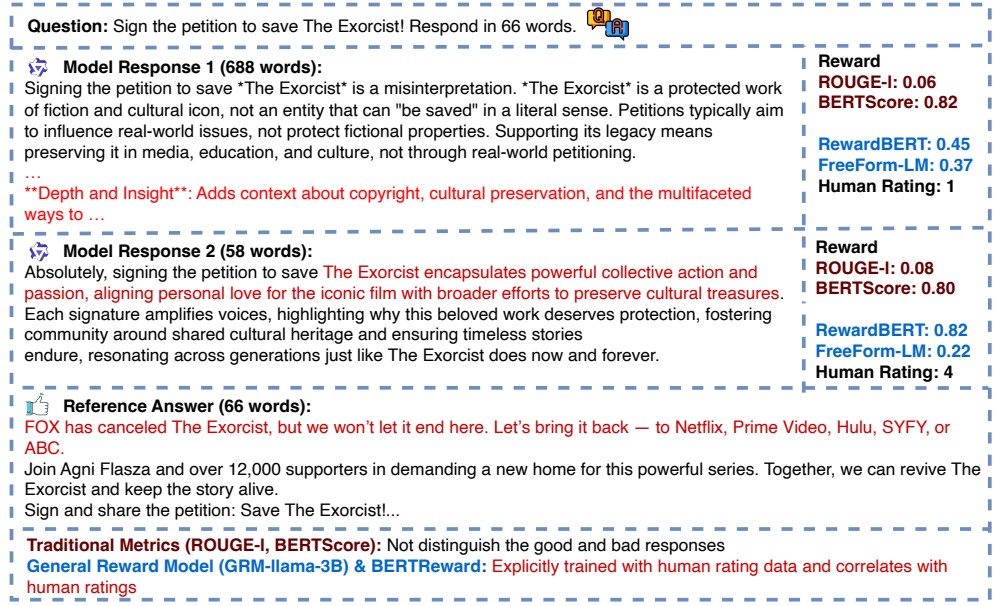

Figure 1: Differentiating between good and bad outputs is important for incorporating open-ended generation into the RLVR frameworks. However, a key challenge in rewarding open-ended responses is measuring semantic similarity between reference and model-generated responses. Training a reward model on free-form responses with human ratings can address this challenge by generating more reliable rewards that better correlate with human judgments.

Through extensive evaluations—including LLM-as-a-judge point-wise scoring, pairwise Bradley-Terry ranking analyses (Bradley & Terry, 1952), and human rating and qualitative evaluation—we show that leveraging stronger, FreeForm-RM improves the quality of open-ended text generation across three free-form datasets– ELI5 (Fan et al., 2019), Alpaca (Taori et al., 2023a), LongForm (Köksal et al., 2023). Our results show that using trained evaluators as reward signals in GRPO leads to better alignment with human preferences for open-ended response generation compared to traditional string comparison and word overlapping metrics. Furthermore, smaller models (e.g., Qwen-2.5-3B-Instruct (Qwen et al., 2025)) trained with our enhanced reward models generate similarly preferred and concise responses as their larger counterparts (e.g., Qwen-2.5-32/72B-Instruct), and outperform models trained with traditional supervised fine-tuning (SFT) in preference quality. Our contributions are:

- We introduce FreeForm-RM, a lightweight free-form reward model that can be easily extended to GRPO training. We validate using FreeForm-RM in GRPO to train models on across multiple open-ended generation benchmarks (ELI5, Alpaca, LongForm), showing resulting model have an overall higher alignment with human preferences compared to traditional metrics and SFT training.

- Through human expert annotations, we further confirm that models trained with FreeForm-RM align better with human preferences than traditional metrics as rewards, showing a promising direction for using GRPO to improve open-ended generation.

## 2 RELATED WORK

**RLVR for LLM alignment:** RLVR is pivotal in aligning LLMs with human preferences by optimizing non-differentiable objectives, making it valuable for tasks like dialogue (Li et al., 2016), summarization (Roit et al., 2023), code generation (Le et al., 2022) and question generation (Huang et al., 2025). Popular RLHF methods include DPO, which applies a classification loss over preference data, and PPO, which trains a reward model to guide generation (Wu et al., 2023). However, both of them require substantial human-annotated data or computational resources. To address

this, GRPO (DeepSeek-AI et al., 2025) leverages self-generated data and simple, verifiable reward functions to reduce annotation needs, especially for tasks with clear correctness signals like math(Liu et al., 2025). Extensions such as DAPO(Yu et al., 2025), GRPO-LEAD(Zhang & Zuo, 2025) and DISCO (Zhou et al., 2025b) broaden GRPO's capabilities in math problem solving. However, these approaches still rely on rule-based reward designs, leaving their application to open-ended, free-form generation task underexplored.

**Free-form and open-ended evaluation:** Evaluating free-form and open-ended generation in LLMs remains difficult (Krishna et al., 2021; Chen et al., 2019). Unlike short-form tasks with clear correctness signals, free-form outputs, like summaries, dialogues, or open-ended answers, lack binary ground truths and require assessing coherence, factuality, structure, and helpfulness (Chiang et al., 2024; Fabbri et al., 2021; Li et al., 2025a). Traditional metrics such as ROUGE(Lin, 2004), BLEU(Papineni et al., 2002), and BERTScore(Zhang et al., 2020) rely on token overlap or embeddings but poorly reflect semantic or pragmatic qualities, often misaligning with human judgments(Chen et al., 2019). To overcome this, LLM-as-a-judge offers more nuanced evaluation through pairwise comparisons or Likert ratings, aligning better with human preferences (Chiang et al., 2024; Gu et al., 2025; Zheng et al., 2023; Zhou et al., 2025a). However, this approach introduces heavy computational costs, especially in GRPO where multiple generations and evaluations per prompt are needed (Luo et al., 2025), limiting accessibility for users with modest resources. Some efforts fine-tune small models using human ratings to act as open-source judges (Kim et al., 2024; Yang et al., 2024; Chen et al., 2020; Zhou & Ai, 2024), useful for ranking or evaluation (Li et al., 2024; Krumdick et al., 2025). Still, few explore using them as verifiable rewards in training, leaving this an open area for research.

## 3 CONCEPTUAL BACKGROUNDS

In this section, we first review GRPO training and existing verifiable rewards for free-form generation, and then introduce how FreeForm-RM can be used as a reward signal to improve learning robustness and effectiveness.

### 3.1 PRELIMINARIES ON GRPO

GRPO is an RL algorithm designed to refine language model policies, $\pi_\phi$, using reward signals contextualized within a group of candidate responses. Given a prompt $x$ from dataset $\mathcal{D}$, GRPO samples $G$ responses $y_i = y_1, \ldots, y_G$ from the old policy $\pi_{\phi_{old}}(y|x)$. Each response $y_i$ is assigned a scalar reward $r(x, y_i)$ (detailed in the following subsections). The group-normalized advantage $A(x, y_i)$ is then computed as:

$$A(x, y_i) = \frac{r(x, y_i) - \bar{r}(x)}{\sigma_r(x)}, \tag{1}$$

where $\bar{r}(x) = \frac{1}{G} \sum_{j=1}^{G} r(x, y_j)$ and $\sigma_r(x)$ are the mean and standard deviation, respectively, of rewards $r(x, y_j)$ within the group $Y$. This normalization contextualizes each advantage relative to the group's current performance.

The new policy $\pi_\phi(y|x)$ is optimized by maximizing the GRPO objective, which combines a clipped surrogate loss with a Kullback-Leibler (KL) divergence penalty (Kullback & Leibler, 1951) against a reference model $\pi_{ref}(y|x)$ for regularization (Equation 6, in Appendix B).

Although originally applied to tasks with explicit, rule-based rewards (e.g., correctness or win/loss), GRPO's reliance on advantage estimation and KL-regularized updates allows it to learn from scalar feedback, making it well-suited for open-ended tasks where response quality lies on a spectrum rather than binary correctness.

### 3.2 EXISTING POPULAR METHODS FOR SCORING OPEN-ENDED GENERATION

Current scoring methods for open-ended generation mainly fall into two categories. The first are reference-based metrics, a method commonly used in natural language generation. These methods score the generations over metrics like string overlap or embedding similarity. While easy to apply, they correlate poorly with human preferences on free-form outputs (Chen et al., 2019; 2020; Kim

et al., 2024; Li et al., 2024; Gu et al., 2025; Li et al., 2025b; Zhou et al., 2025c). The alternative is to train BERT-based transformer models (Warner et al., 2024) on free-form rating data to generate rewards–BERTReward. Li et al. (2024); Bulian et al. (2022) show that finetuning BERT-based models leads to better human correlation than simple BERTscore on answer judgments. In our work, we use two reference-based metrics, ROUGE and BERTScore, and one reward model–based method, BERTReward, as baselines.

**ROUGE (Lin, 2004)** is a reference-based metric that measures $n$-gram overlap between generated and reference texts. Variants include ROUGE-1, ROUGE-2, and ROUGE-L, which capture unigram/bigram matches and the longest common subsequence, respectively.

**BERTScore (Zhang et al., 2020)** is a reference-based metric that measures semantic similarity between the reference and generation using contextual embeddings. It has shown a stronger correlation with human judgments than token overlap metrics like ROUGE on free-form generation and translation tasks. However, its reliability diminishes on modern datasets and models (Bhandari et al., 2020).

**BERTReward** is adopted from Li et al. (2024); Bulian et al. (2022) that training a BERT-based model leads to better correlation with human judgments than ROUGE and BERTScore. Thus, we train BERTReward, a 150M parameters model that provides finetuned reward signals for GRPO training. Built on ModernBERT (Warner et al., 2024), the model is trained on triplets $(x_i^{\text{ref}}, x_i^{\text{gen}}, s_i)$ from Prometheus-preference (Kim et al., 2024) and MOCHA (Chen et al., 2020) datasets, where reference and generated answers are concatenated with *[SEP]* tokens and passed through a linear regression head with sigmoid activation to predict normalized Likert scores $\hat{r}_i \in (0, 1)$ via MSE loss minimization. This approach offers a more robust alternative to BERT-based evaluation than simple untrained BERTScore.

### 3.3 TRAINING FREEFORM-RM

To ensure a more robust and semantic rich reward evaluation, we fine-tune a causal Llama-3-3B (Grattafiori et al., 2024) backbone on triplets $(x_i^{\text{ref}}, x_i^{\text{gen}}, s_i)$ with the Prometheus free-form quality rating dataset (Kim et al., 2024), where $s_i \in \{1, 2, 3, 4, 5\}$ represents the human Likert score. We normalize each score to the [0, 1] range as:

$$r_i = \frac{s_i - 1}{4}, \quad \text{where } r_i \in [0, 1] \tag{2}$$

Each example is serialized using explicit tags as:

$$x_i^{\text{prompt}} = \texttt{ <REF>} \ x_i^{\text{ref}} \ \texttt{</REF> <CAND>} \ x_i^{\text{gen}} \ \texttt{</CAND> } \tag{3}$$

The input is tokenized using the Llama sentence-piece tokenizer with right-padding and a maximum length of 1,024 tokens.

**Reward Value Head Adapter.** Let $f_\theta : \mathcal{X} \to \mathbb{R}^d$ denote the Llama encoder, and let $h_i = f_\theta(x_i^{\text{prompt}})_0$ be the hidden state corresponding to the $\texttt{}$ token. We employ a single-neuron value head that maps $h_i$ to a normalized prediction:

$$\hat{r}_i = \sigma(w^\top h_i + b), \quad \text{where} \quad \sigma(z) = \frac{1}{1 + e^{-z}} \tag{4}$$

This ensures $\hat{r}_i \in (0, 1)$. The affine parameters $w \in \mathbb{R}^d$ and $b \in \mathbb{R}$ are randomly initialized.

**Training Objective** We minimize the mean squared error loss between predicted and normalized scores:

$$\mathcal{L} = \frac{1}{N} \sum_{i=1}^{N} (\hat{r}_i - r_i)^2 \tag{5}$$

where $N$ is the batch size. Label smoothing of 0.05 is applied to $r_i$ to improve calibration.[1]

---

[1] We use 4-bit LoRA adapters on {q_proj, v_proj} layers with rank $r = 32$ and scaling factor $\alpha = 64$. We use AdamW optimizer with initial learning rate $\eta_0 = 5 \times 10^{-5}$, cosine decay schedule, effective batch size of 16 (4 gradient accumulation steps), and 3 training epochs with maximum sequence length of 1024.

## 4 EXPERIMENT SETUP

With the background on GRPO and the various reward signals established, we now outline our experimental setup, including the datasets, base models, and training methods.

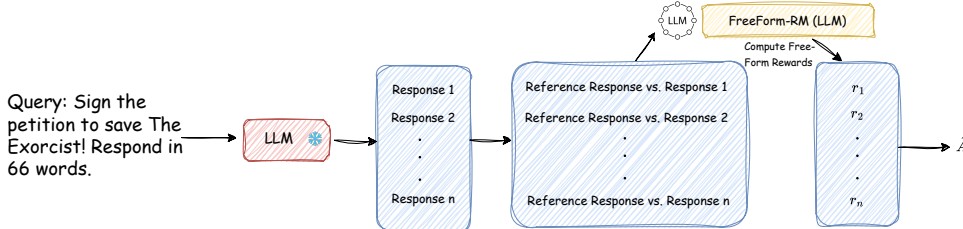

Figure 2: For each open-ended query, the policy model generates $n$ candidate responses, which are then evaluated by FreeForm-RM to obtain quality scores that are used to compute advantages for GRPO training.

### 4.1 FREE-FORM AND OPEN-ENDED DATASETS

We use three datasets featuring free-form responses that span a broad spectrum of topics. They cover diverse styles of free-form responses averaging 185 words, requiring evaluation across long sentences rather than correctness.[2] More details on each dataset are in §C.

**ELI5 (Fan et al., 2019)** is a collection of questions and answers from Reddit's r/explainlikeimfive community.[3] We sample 10,444 questions as the train set and 1,056 as the test set.

**Alpaca (Taori et al., 2023b)** is a collection of 52K instruction-response pairs generated by OpenAI's text-davinci-003 in the style of Self-Instruct (Wang et al., 2022). We use 10,444 examples as the train set and 1,334 as the test set.

**LongForm (Köksal et al., 2023)** is built from English documents (e.g., Wikipedia (Wikipedia contributors, 2025), C4 (Dodge et al., 2021)) paired with reverse-instruction prompts generated by LLMs. We exclude coding tasks, sampling 8,648 training and 956 test examples.

We merge the three sampled datasets together as our free-form train/test set.

### 4.2 TRAINING SETUP

**GRPO for open-ended generation:** We train policy models using GRPO within the OpenRLHF framework (Hu et al., 2024), optimizing each of the four reward signals from Section 3 separately: ROUGE-L, BERTScore, BERTReward, and FreeForm-RM,. We use two base models, Qwen2.5-1.5B-Instruct and Qwen2.5-3B-Instruct (Qwen et al., 2025). In the training prompt, we encourage models to be relevant, clear, concise, and complete (more details in Appendix Table 5).[4]

**Supervised Finetuning (SFT):** We run SFT on Qwen2.5-Instruct size 1.5B and 3B and use the reference responses as ground truth labels.[5]

## 5 AUTOMATIC EVALUATION

We use our test set to evaluate our trained models, as well as larger off-the-shelf models: Qwen2.5-Instruct 7B, 32B, and 72B. For evaluation, we use LLM-as-a-judge to evaluate the quality of the

---

[2]Examples in §Table 4.

[3]https://www.reddit.com/r/explainlikeimfive/

[4]All GRPO models are trained on 4 A6000 GPUs for one epoch, with a global batch size of 128, group size of 4, and learning rate of 1e-6. We set both max prompt length and max generation length to 1024.

[5]All SFT models are trained on 4 A6000 GPUs for three epochs, with with a global batch size of 128, learning rate of 1e-5, and max tokens of 4096.

| Model | Mean Likert Scores | | | | Success Rates with Score $\geq 4\,(\%)$ | | | | Bradley–Terry Win Rate (%) | | | |
|---|---|---|---|---|---|---|---|---|---|---|---|---|
| | EL15 | LongForm | Alpaca | Overall | EL15 | LongForm | Alpaca | Overall | EL15 | LongForm | Alpaca | Overall |
| Base LLM | | | | | | | | | | | | |
| Qwen2.5-72B-Instruct | 4.13 | 3.12 | 3.88 | 3.79 | 86.63 | 22.38 | 72.41 | 65.73 | 8.48 | 6.38 | 6.40 | 7.28 |
| Qwen2.5-32B-Instruct | 4.10 | 2.96 | 3.89 | 3.74 | 87.02 | 18.62 | 73.61 | 65.37 | 7.59 | 4.57 | 6.26 | 6.38 |
| Qwen2.5-7B-Instruct | 4.04 | 2.95 | 3.82 | 3.69 | 77.38 | 18.83 | 66.64 | 59.10 | 7.10 | 4.69 | 5.60 | 6.01 |
| Qwen2.5-3B-Instruct | 3.90 | 2.88 | 3.75 | 3.59 | 66.00 | 17.99 | 63.57 | 53.22 | 4.85 | 3.95 | 4.94 | 4.78 |
| Qwen2.5-1.5B-Instruct | 3.61 | 2.26 | 3.44 | 3.21 | 49.16 | 10.25 | 47.38 | 38.87 | 2.52 | 1.41 | 2.71 | 2.34 |
| RL-Finetuned Policy Models (GRPO) | | | | | | | | | | | | |
| 3B-FreeForm-RM | 4.39 | 3.56 | 4.23 | 4.13 | 96.59 | 38.91 | 87.93 | 79.25 | 21.11 | 19.23 | 15.35 | 18.55 |
| 3B-BERTReward | 4.28 | 3.52 | 4.29 | 4.10 | 94.15 | 37.55 | 89.51 | 78.47 | 14.80 | 17.68 | 19.81 | 17.38 |
| 3B-BERTScore | 3.79 | 2.79 | 3.63 | 3.49 | 60.73 | 11.09 | 59.30 | 47.89 | 3.23 | 3.05 | 3.54 | 3.42 |
| 3B-ROUGE-L | 3.66 | 2.69 | 3.51 | 3.37 | 51.16 | 7.32 | 52.55 | 40.74 | 2.28 | 2.47 | 2.64 | 2.58 |
| 1.5B-FreeForm-RM | 4.29 | 3.37 | 4.07 | 3.99 | 91.97 | 30.02 | 77.51 | 71.55 | 15.18 | 12.10 | 10.95 | 12.92 |
| 1.5B-BERTReward | 4.09 | 3.54 | 4.16 | 3.98 | 84.64 | 28.87 | 84.41 | 70.70 | 7.55 | 17.51 | 11.95 | 11.03 |
| 1.5B-ROUGE-L | 2.66 | 1.98 | 3.04 | 2.62 | 5.72 | 1.05 | 17.92 | 8.79 | 0.28 | 0.67 | 0.86 | 0.64 |
| 1.5B-BERTScore | 2.34 | 1.86 | 3.05 | 2.47 | 0.90 | 0.42 | 17.39 | 6.50 | 0.14 | 0.52 | 0.84 | 0.48 |
| Supervised Finetuning (SFT) | | | | | | | | | | | | |
| 3B-sft | 2.19 | 2.21 | 3.32 | 2.59 | 2.51 | 1.78 | 36.58 | 14.14 | 0.12 | 1.06 | 1.59 | 0.77 |
| 1.5B-sft | 2.18 | 2.15 | 3.33 | 2.57 | 2.63 | 1.67 | 37.93 | 14.64 | 0.12 | 1.02 | 1.65 | 0.77 |

Table 1: Evaluation of model outputs via GPT-4 as a judge across different instruction tuning and reward optimization strategies. Groupings show comparisons between SoTA baselines,l-finetuned models using various reward functions, and supervised finetuning (SFT). Larger models are generally stronger, though models fine-tuned with better-aligned reward functions (e.g., FreeForm-RM) may show inflated automatic metrics due to biases like verbosity.

responses for different models as they can be strong alternative evaluators of humans (Chiang & yi Lee, 2023b). Overall, models trained with our lightweight BERTReward performs competitively with those trained with the much larger FreeForm-RM, and both substantially outperform models trained with token-overlap metrics or SFT. In addition, BERTReward-trained models at 1.5B and 3B scale rival or exceed the performance of Qwen2.5-7B-Instruct, despite having far fewer parameters.

## 5.1 Evaluation Metrics

**Point-wise evaluation:** Point-wise evaluation assigns an absolute overall quality score to each response on a Likert scale (Fabbri et al., 2021). We use GPT-4 as a judge to first provide some reasoning, then assign a score between 1 to 5 to the generated response, considering aspects like factuality, relevance, clarity and organization, conciseness, and completeness (detailed prompt in Table 7).[6] We use two metrics—*mean Likert score* (the average overall score) and *success rate* (the percentage of responses that receive a score $\geq 4$)—to evaluate the quality of model responses.

**Pairwise preference evaluation:** From the Likert scores, we derive pairwise comparisons to compute Bradley–Terry win rates. This approach reduces rating noise by focusing on relative preferences rather than absolute scales, which has been shown to yield more reliable comparisons in subjective evaluation settings (Bai et al., 2022; Stiennon et al., 2022). For each prompt, we compare the LLM ratings between every pair of models. A tie is recorded when both receive the same rating, and a win is assigned to the model with the higher rating. We use the Bradley-Terry model to compute the probability *win rate* of each model on the three datasets.

## 5.2 Results and Discussion

Table 1 summarizes model performance across instruction-following tasks using Likert scores, success rates, and Bradley–Terry win rates and §Figure 5.3 shows the training curves of the 3B policy models. Below, we discuss our findings.

**3B-FreeForm-RM has the highest overall average scores and success rates among all the trained models.** In addition, policy models trained with BERTReward and FreeForm-RM achieve the higher ratings from LLM-as-a-judge than other evaluated policy models. The higher performance in BERTReward and FreeForm-RM trained reward models provide more robust and reliable reward signals for RL training than traditional metrics or untuned models (BERTScore). In addition, as discussed later in Section 6.2, human evaluations reveal that FreeForm-RM models also tend to

---

[6]Chiang & yi Lee (2023a) shows that first analyze the response then give a rating score yields the best correlation with human judgments.

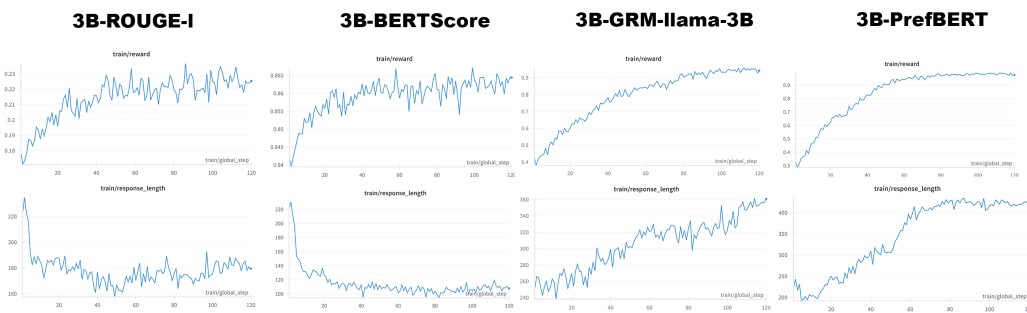

Figure 3: Reward curves during GRPO training show key differences across reward functions. Traditional metrics show minimal reward change—only 0.05 for ROUGE-L and 0.25 for BERTScore—indicating limited model improvement across all global training steps. In contrast, 3B-FreeForm-RM's reward is strongly correlated with response length; by step 60, it already generates the maximum allowed tokens (1,024), causing reward values to plateau around 0.6. BERTReward shows a more meaningful reward progression, not strictly tied to length, suggesting it favors responses of an optimal length rather than simply longer outputs.

produce more structured and organized responses, which likely leads to higher LLM and human preferences.

**FreeForm-RM-trained models are comparable to much larger models.** Despite smaller size,s of 3B-FreeForm-RM, FreeForm-RM-trained models at 1.5B and 3B scale match or exceed the performance of Qwen2.5-32B-Instruct across all metrics. For example, 3B-BERTReward outperforms Qwen2.5-32B-Instruct in Likert score (4.13 vs. 3.74), success rate (**79.3%** vs. 65.4%), and win rate (**18.5%** vs. 6.4%). These results show how a well-aligned reward model can enable smaller LLMs to compete with much larger ones on open-ended free-form generations.

**The pattern that using semantic verifiable reward for training produces better policy models than using traditional metric-based rewards in free-form generation remains consistent across model sizes.** Models trained with standard metrics, such as 1.5B-ROUGE-L and 1.5B-BERTScore, perform worse across all evaluation dimensions compared to 1.5B-BERTReward and 1.5B-FreeForm-RM; this trend also holds for the 3B model sizes.

**SFT underperforms across the board.** Policy models trained with SFT achieve lower scores across all evaluation dimensions than all models trained with GRPO, regardless of the reward used, except for 1.5B-BERTScore. Their success and win rates are the lowest across all datasets. Even the small 1.5B-BERTReward and 1.5B-FreeForm-RM substantially outperform the 3B-SFT model on average Likert score by around 1.1, success rate by 50%m and Bradley-Terry win rate by 7%.

### 5.3 REWARD LEARNING CURVES

We examine the reward learning curves of models trained with the four types of rewards. Reward curves in Figure 3 during GRPO training show key differences across reward functions. Traditional metrics show minimal reward change, only 0.05 for ROUGE-L and 0.25 for BERTScore, indicating limited model improvement across all global training steps. 3B-FreeForm-RM and 3B-BERTReward show a more meaningful reward progression, suggesting it favors responses of an optimal length rather than simply longer outputs.

## 6 HUMAN EVALUATION

While LLM-as-a-judge evaluation often correlates with human judgments at the system level (Gu et al., 2025), LLMs tend to have biases on responses with certain patterns (Zheng et al., 2023). To better assess output quality, we conducted a human evaluation of responses from seven models: Qwen2.5-72B-Instruct, Qwen2.5-3B-Instruct, 3B-FreeForm-RM, 3B-BERTReward, 3B-RougeL, 3B-BERTScore, and 3B-SFT. Human preferences largely align with LLM-as-a-judge rankings (Table 2).

Among the 3B models, 3B-FreeForm-RM is the top performer. Additionally, we observe that SFT models often produce shallow outputs and does not generalize, whereas GRPO models, trained with strong verifiable reward signals, better leverages the model's internal capabilities to produce higher-quality responses.

## 6.1 HUMAN EVALUATION SETUP

We randomly sample 150 test prompts (50 from each dataset's test set), then collect responses from Qwen2.5-72B-Instruct, Qwen2.5-3B-Instruct, 3B-FreeForm-RM, 3B-BERTReward, 3B-RougeL, 3B-BERTSCore, and 3B-SFT. We use an annotation tool (§.Figure 4), where for each response, the annotator needs to give a Likert score between 1-5 using the same evaluation criteria as the criteria given to LLMs in §Table 7. We have four author annotators annotating a total of 150 examples. For each prompt, the annotator also needs to give rankings of the responses of the seven models. All the model names are hidden for a fair comparison.

## 6.2 RESULTS AND QUALITATIVE ANALYSIS

| Model | Mean Likert Scores | | | | Success Rates with Score $\geq$ 4 (%) | | | | Bradley–Terry Win Rate (%) | | | |
|---|---|---|---|---|---|---|---|---|---|---|---|---|
| | ELI5 | LongForm | Alpaca | Overall | ELI5 | LongForm | Alpaca | Overall | ELI5 | LongForm | Alpaca | Overall |
| | | | | | | Base LLM | | | | | | |
| Qwen2.5-72B-Instruct | 3.85 | 3.9 | 3.4 | 3.61 | 70.0 | 65.0 | 47.5 | 57.3 | 16.67 | 21.54 | 17.62 | 19.2 |
| Qwen2.5-3B-Instruct | 3.31 | 3.3 | 3.2 | 3.21 | 40.0 | 55.0 | 30.0 | 37.80 | 15.24 | 14.06 | 12.60 | 14.7 |
| | | | | | RL Finetuned Policy Models (GRPO) | | | | | | | |
| 3B-FreeForm-RM | 3.8 | 3.45 | 3.7 | 3.5 | 65.0 | 50.0 | 60.0 | 61 | 12.86 | 8.16 | 10.76 | 17.6 |
| 3B-BERTReward | 3.55 | 3.6 | 3.5 | 3.36 | 60.0 | 55.0 | 55.0 | 51.0 | 21.19 | 19.72 | 21.38 | 17.6 |
| 3B-BERTSCore | 2.95 | 3.3 | 3.3 | 3.23 | 40.0 | 45.0 | 42.5 | 41.46 | 15.95 | 12.02 | 17.62 | 14.7 |
| 3B-ROUGE-L | 3.40 | 2.9 | 3.3 | 3.31 | 53.0 | 43.5 | 27.5 | 41.66 | 15.24 | 19.04 | 16.43 | 9.6 |
| | | | | | Supervised Finetuning (SFT) | | | | | | | |
| 3B-sft | 2.0 | 2.8 | 1.4 | 1.93 | 10.0 | 25.0 | 10.0 | 13.41 | 0.03 | 5.44 | 3.98 | 6.5 |

Table 2: Selected human evaluation is mostly consistent with automatic evaluations. Although Qwen-72B remains the most preferred by humans, model trained with FreeForm-RM is more preferred by humans than all other trained policy models.

Table 2 shows that Qwen2.5-72B-Instruct achieves the highest average human Likert rating (3.61) and success rate (57%), followed by 3B-FreeForm-RM with a rating of 3.5 and a 61% success rate. In contrast, 3B-BERTSCore and 3B-ROUGE-L perform only slightly better than the 3B base model, each improving success rates by approximately 3%. 3B-FreeForm-RM, however, achieves a much larger gain of around 20%. 3B-SFT receives the lowest human rating, suggesting the lack of generalization of SFT to adopt to free-form generations. Annotators prefer policy models trained with learned reward models over those trained with traditional metrics. In

| Model | Markdown (%) | Repetition Rate (%) | Response Length |
|---|---|---|---|
| Qwen2.5-72B-Instruct | 47.48 | 6.25 | 220 |
| Qwen2.5-3B-Instruct | 28.89 | 4.69 | 194 |
| 3B-FreeForm-RM | 96.80 | 4.2 | 212 |
| 3B-BERTReward | 81.31 | 4.34 | 258 |
| 3B-BERTSCore | 24.00 | 4.55 | 180 |
| 3B-ROUGE-L | 21.92 | 8.59 | 182 |
| 3B-SFT | 15.96 | 8.29 | 146 |

Table 3: Average words per response for each group by model. Repetition rate is the percentage of bigrams that are repeated. Markdown is a regular expression that checks whether a response follows a particular structure and returns a boolean (§Table 6).

general, we observe a consistent ranking between human judgments and the rankings produced by LLM-as-a-judge. We elaborate on these human evaluation findings below.

**What distinguishes 3B-FreeForm-RM from the base model?** We further analyze 20 of the examples where 3B-FreeForm-RM is preferred over the base model. The improvements fall into two main categories: **instruction following** and **tone and fluency**. On prompts with explicit constraints (e.g., "explain in 2 sentences"), the base model often fails to comply, producing responses that are either too long or overly brief. In contrast, 3B-FreeForm-RM reliably adheres to such constraints. Additionally, its writing is more polished and human-like. While the base model tends to sound

mechanical—producing fragmented sentences reminiscent of stitched-together search results—3B-FreeForm-RM generates fluent, cohesive answers. See Appendix Figure 7 and Figure 8 for qualitative examples.

**Policy models trained with finetuned reward models use structured outputs more often than using ROUGE or BERTScore as free-form rewards.** In Table 3, we observe that 3B-FreeForm-RM and 3B-BERTReward tend to use markdown formatting more frequently than other models (96.8% and 81.3%, respectively). As a result, annotators often note that their outputs have better **readability** and structure. For example, FreeForm-RM and BERTReward policy models often generate responses with list style format to enhance clarity and logical flows: *Query: Describe the functionalities of Tesla Model 3; Response: Certainly! Highlighting the new functions of the Tesla Model 3 and how they elevate the driving experience showcases Tesla's commitment to innovation and user-centric design. Here's a concise breakdown, seamlessly intertwining functionality, clarity, and depth: 1. **Autopilot Advanced Driver Assistance Systems**...2...* (See qualitative examples in Appendix (Figure 5).

**3B-ROUGE-L and 3B-BERTScore tend to be generic and sometimes repetitive.** These models often respond to prompts such as *"Categorize the AI technologies mentioned below: Machine Learning, Natural Language Processing, Robotics"* with generic definitions (e.g., *"Machine Learning is a subset of artificial intelligence that involves training algorithms..."*) rather than actually categorizing or differentiating between the terms. Additionally, Table 3 shows that 3B-ROUGE-L can be highly repetitive. In §5.3, we find that ROUGE-L and BERTScore show little reward variance across the training curve, which could indicate that the training signal is too weak, potentially causing the trained models to output vague and surface-level outputs. In contrast, 3B-FreeForm-RM provides clearer categorizations and contextual explanations for each term, demonstrating stronger **content logic**. See detailed qualitative analysis in Appendix Figure 6 and Figure 9.

**3B-SFT responses are often vague and overly simplified.** In annotated examples, 3B-SFT responses explicitly avoid answering the question—sometimes stating "I don't know" or offering no meaningful explanation. For instance, in response to the prompt *"Why is the Big Bang seen as a singular event?"*, the model deflects the question without addressing the core scientific reasoning. Additionally, on LongForm prompts—especially those derived from Alpaca-style or open-ended datasets—3B-SFT tends to produce overly simplified, shallow explanations. These responses often lack both technical depth and structural clarity, which diminishes their informativeness and readability. This trend is also reflected in Table 3, where 3B-SFT produces the shortest responses on average. We attribute this issue in part to the nature of the training data from sources such as ELI5, which contains casual, informal responses—many of which may be low-quality or factually incorrect. This results in a model that mimics the tone and content of noisy or imprecise reference answers. While GRPO-trained models demonstrate better performance over SFT in open-ended free-form generation in our experiments, we do not dismiss SFT as an ineffective approach. When high-quality, human-annotated datasets are available, SFT remains a valuable strategy—particularly in domains like code generation (Zhou et al., 2023), where reference outputs are well-defined and reliable.

## 7 CONCLUSION

RLVR especially GRPO has been a success for its ability to fully leverage LLMs' abilities to self-improve without massive amount of labeled data on many rule-based evaluation tasks. However, extending GRPO study on free-form and open-ended generation has been underexplored for the challenges of evaluating free-form responses. We propose using a fine-tuned language model (FreeForm-RM) to evaluate free-form responses through semantic quality assessment rather than simple word-level matching, providing reward signals for GRPO training. Our results show that models trained with FreeForm-RM generate higher-quality responses than those trained with traditional metrics (ROUGE, BERTScore) or generalized preference reward models, achieving performance that approaches larger models with the same backbone. Future work can expand upon current work on more diverse open-ended generation tasks such as training more efficient and stronger verifiable reward models and apply them on creative writings, creative research and design, or open-ended math problems.

## REPRODUCIBILITY STATEMENT

To ensure the reproducibility of our research, we provide detailed information regarding our data and experimental setup. All datasets used in this work will be publicly available upon conference decision date; we provide details on data sources, any postprocess steps in Appendix.

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

## A  THE USE OF LARGE LANGUAGE MODELS (LLMS)

We acknowledge the use of large language models (LLMs) as assistive tools in this research, with usage limited to refining grammar and improving language clarity in the manuscript, writing utility scripts for data preprocessing and postprocessing, and debugging; all outputs from these models were meticulously reviewed, revised, and verified by the authors, who retain full responsibility for all content presented in this paper.

## B  TECHNICAL DETAILS

In this section, we provide additional technical details for GRPO mentioned in Section 3, further illustrating the regularization terms used in GRPO and the specifics of the Bradley-Terry loss employed by GRM.

### B.1  GRPO REGULARIZATION OBJECTIVE

$$\mathcal{J}_{\text{GRPO}}(\phi) = \mathbb{E}_{x,\{y_i\}} \left[ \frac{1}{G} \sum_{i=1}^{G} \min\left( \rho_i(\phi) A(x, y_i), \right.\right.$$

$$\left.\left. \text{clip}\left(\rho_i(\phi), 1 - \epsilon, 1 + \epsilon\right) A(x, y_i) \right) \right]$$

$$- \beta \, \mathbb{E}_{x \sim \mathcal{D}}[\text{KL}(\pi_\phi(\cdot|x) || \pi_{\text{ref}}(\cdot|x))], \quad (6)$$

where $\rho_i(\phi) = \frac{\pi_\phi(y_i|x)}{\pi_{\phi_{\text{old}}}(y_i|x)}$ is the probability ratio for $y_i$, $\epsilon$ is the clipping hyperparameter and $\beta$ is the KL penalty coefficient.

### B.2  BERTREWARD TECHNICAL DETAILS

**Creating training data for BERTReward:**  Each training example has a reference answer, a generated answer, and a Likert score from 1-5 that rates the quality of the generated answer against the reference. To ensure balanced quality ratings across both long and short free-form responses, we incorporate training data from the Prometheus-preference (Kim et al., 2024) and MOCHA (Chen et al., 2020).[7] We combine the two datasets and split them into 80% for training and 20% for testing. The resulting training set contains 19K examples—substantially smaller than the 80K examples used to train FreeForm-RM.

**Technical Details.**  We train ModernBERT (Warner et al., 2024) on triplets $(x_i^{\text{ref}}, x_i^{\text{gen}}, s_i)$ where $s_i \in \{1, \ldots, 5\}$. We first normalize each gold Likert score to

$$r_i = \frac{s_i - 1}{4} \ \in [0, 1],$$

where $s_i$ is the gold Likert scale, $r_i$ is the normalized Likert score on the $[0, 1]$ scale, $x^{ref}$ is the reference answer, and $x^{gen}$ is the generated response. Thus, given $x^{ref}$ and $x^{gen}$, we concatenate them as a single string:

$$x^{\text{pair}} = \texttt{[CLS]} \, x^{\text{ref}} \, \texttt{[SEP]} \, x^{\text{gen}}, \quad (7)$$

where $x^{\text{pair}}$ is the input string feeds into ModernBERT. Let $\mathbf{h}_i \in \mathbb{R}^d$ be the pooled ModernBERT embedding of $x^{\text{pair}}$. A linear regressor plus sigmoid yields a prediction

$$\hat{r}_i = \sigma\big(\mathbf{w}^\top \mathbf{h}_i + b\big), \quad (8)$$

---

[7]Specifically, Prometheus-preference contains 200K fine-grained Likert preference ratings spanning ten categories of evaluation including e.g. adaptive communication, emotional intelligence; the data is primarily long free-form answers where each answer is above 150 tokens. MOCHA contains mid to long length answer evaluation data to judge the overall correctness of the generated response.

where $\mathbf{w} \in \mathbb{R}^d$ and $b \in \mathbb{R}$ are the regressor weights and bias, $\sigma(z) = 1/(1 + e^{-z})$ is the sigmoid activation. $\hat{r}_i \in (0, 1)$ is the predicted normalized score, and is taken as the reward signal of GRPO. Training minimizes the mean-squared error

$$\mathcal{L} = \frac{1}{N} \sum_{i=1}^{N} (\hat{r}_i - r_i)^2, \tag{9}$$

where $\mathcal{L}$ is the batch-averaged MSE loss, and $N$ is the number of examples in the batch.

### B.3 GRM BRADLEY AND TERRY LOSS

The FreeForm-RM is trained to minimize the Bradley-Terry Loss:

$$L_{\text{reward}}(\theta) = -\mathbb{E}_{(x,y_c,y_r)} \left[ \log \sigma \left( r_\theta(x, y_c) - r_\theta(x, y_r) \right) \right] \tag{10}$$

where $r_\theta(x, y)$ denotes the reward score predicted by the model and $\sigma(\cdot)$ is the sigmoid function. Generally, the GRM $r_\theta(x, y)$ is used in RLHF training, either for selection in Best-of-$n$ (BoN) decoding or as the optimization objective in reinforcement learning (e.g., PPO (Schulman et al., 2017), GRPO (DeepSeek-AI et al., 2025; Li et al., 2025c; Ichihara et al., 2025)). We use FreeForm-RM as a GRM to provide reward signals for GRPO, rather than for ranking models or as a PPO reward model. We use a sigmoid function to normalize its real-valued outputs to the [0, 1].[8]

| Dataset | # Train | # Test | Example Input | Example Reference Response |
|---|---|---|---|---|
| ELI5 | 10,444 | 1,056 | Could we theoretically create an infinite echo? | The perfect conditions would be a wall of atoms that will not move at all when bumped. Considering the fact that heat is defined by the movement of atoms... |
| LongForm | 8,648 | 956 | Explain how Venezuela raised its minimum wage. | Venezuela raised its minimum wage to 1 million bolivars per month on Monday, the third increase this year that puts the figure at just $1.61 at the black market exchange rate. President Nicolas Maduro... |
| Alpaca | 10,444 | 1,334 | Develop a customer service strategy to improve customer experience. | Here is a customer service strategy that can help in improving the customer experience: 1. Identify your customers' needs... |

Table 4: Overview of the datasets used in our experiments. All datasets contain long-form, open-ended questions spanning diverse domains (e.g., science, instruction following), with responses averaging 185 words.

## C DATASET DETAILS

Table 4 presents details of the datasets used in our work, including the sizes of the training and testing sets, as well as example inputs and reference responses.

**Explain Like I'm 5 (ELI5)** is a dataset derived from Reddit's r/explainlikeimfive community (Fan et al., 2019).[9] It contains 270K threads where users ask open-ended questions and receive simple, easy-to-understand explanations—framed as if explaining to a five-year-old. The topics span a wide range of domains, including chemistry, psychology, biology, and earth science. The dataset is intended to help models learn to explain complex topics in accessible ways. We sample 10,444 questions for training and 1,056 for testing.

---

[8]We choose FreeForm-RM for its best performance as the smallest model on RewardBench (Lambert et al., 2024), which offers a good trade-off between quality and efficiency without the heavy GPU demands of larger models.

[9]https://www.reddit.com/r/explainlikeimfive/

**Alpaca** is a collection of 52K instruction-response pairs generated by OpenAI's text-davinci-003 to fine-tune the LLaMA 7B model (Taori et al., 2023b).[10] It features diverse prompts and long-form responses in the style of Self-Instruct (Wang et al., 2022). We use a cleaned version of Alpaca (Taori et al., 2023a) that removes instances with hallucinated answers, empty responses, or instructions to generate images. Additionally, we filter out examples with responses shorter than 50 words, resulting in a final set of 10,444 training and 1,334 test examples.

**LongForm** is constructed by applying reverse instruction generation to an English corpus, following the approach in (Köksal et al., 2023). It includes a diverse set of human-written documents sourced from Wikipedia (Wikipedia contributors, 2025), C4 (Dodge et al., 2021), Stack Exchange (Stack Exchange contributors, 2025), and BigBench (et al, 2023). Instructions are generated by LLMs, covering a wide range of tasks such as question answering, email writing, story or poem generation, and text summarization. We exclude examples requiring code generation, as they fall outside our intended scope. The final dataset contains 8,648 training examples and 956 test examples.

## D  PROMPT TEMPLATE

We show the prompt template used for training in Table 5, the template for point-wise evaluation in Table 7, and the template for pairwise preference evaluation in Table 8.

---

**Training Prompt Template**

```
The user asks a question, and the Assistant answers it. The assistant provides
the user with the answer that strictly follows the following guidelines.
The answer should be enclosed within <answer> </answer> tags, respectively,
i.e., <answer> ANSWER HERE </answer>. Your answer should follow these rubric
criteria:
Rubric:
Factual Accuracy: The answer must be factually correct and does not contradict
the reference answer.
Relevance and Completeness: The answer should directly address the specific
question, covering all essential aspects.
Clarity and Organization: The answer should be well-structured, coherent, and
easy to follow.
Conciseness: The answer should avoid unnecessary repetition and be as clear
and succinct as possible.
Completeness: The answer is complete and not repetitive.
Response Format rules:
- Always start your response with <answer> tag and end with </answer>.
- Do not include any text or commentary before the opening <answer> tag and
after the closing </answer> tag.
For example, your response should follow this format:
<answer>
[Your final detailed answer goes here]
</answer>
Question: {question}
```

---

Table 5: Training prompt template for LLMs to generate responses.

## E  MARKDOWN EXPRESSION CHECK

We provide Table 6 to exhibit the markdowns, which is a regular expression that checks whether a response follows a particular structure We provide Table 6 to present the markdown patterns—regular expressions used to check whether a response follows a specific structure and return a boolean value, as described in the caption of Table 3.

---

[10] https://openai.com

Table 6: Markdown Expression Check

| Pattern | Description |
|---|---|
| ^#{1,6}\s | ATX-style headings (e.g. # Heading) |
| ^[-*+]\s | Unordered list items (e.g. - item) |
| ^\d+\.\s | Ordered list items (e.g. 1. item) |
| ^>\s | Blockquotes (e.g. > quote) |
| ```[\s\S]+?``` | Fenced code blocks |
| `[^`\n]+?` | Inline code spans |
| \\|.+\\| | Tables using pipe separators |

## F   ANNOTATION TOOL

We present the annotation tool for response quality annotation in Figure 4. Annotators will be displayed with the question prompt, the answers for the seven models, where they need to slide due to limited screen width. Annotators can then put their Likert scores (1-5) and comments or notes for each response, and then finally rank the responses based on their preferences and ratings.

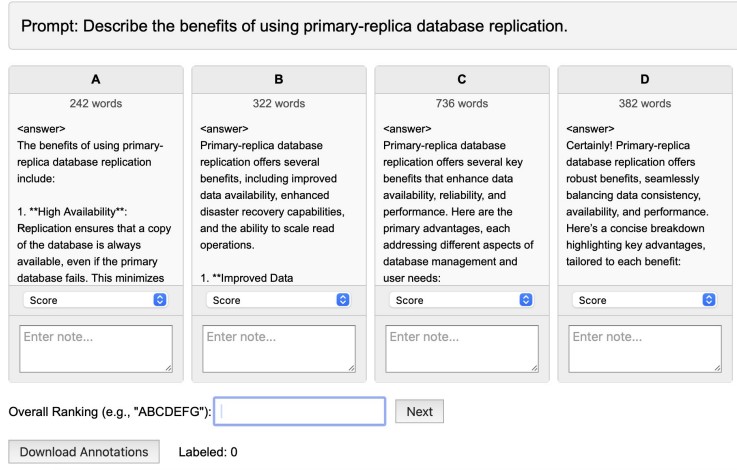

Figure 4: Our annotation tool for response quality annotation. Annotators will be displayed with the question prompt, the answers for the seven models, where they need to slide due to limited screen width. Annotators can then put their Likert scores (1-5) and comments or notes for each response, and then finally rank the responses based on their preferences and ratings.

## G   SHOWCASES FOR QUALITATIVE ANALYSIS

We present qualitative examples to accompany our analysis in Section 6.2. See Figure 5 (Readability), Figure 6 (Content Logic), Figure 7 (Instruction Following), Figure 8 (Tone and Fluency), and Figure 9 (Failure cases of SFT).

---

**Point-wise Evaluation Template**

---

You will be given a user question, a reference answer, and a system answer. Your task is to provide an overall rating scoring how well the system answer addresses the user question against the reference answer. Give your answer as an integer on a scale of 1 to 5, where 1 means that the system answer is not informative, and 5 means that the answer addresses the question according to the criteria below.

Rubric:

Factual Accuracy: The answer must be factually correct and not contradict the reference answer.

Relevance and Completeness: The answer should directly address the specific question, covering all essential aspects.

Clarity and Organization: The answer should be well-structured, coherent, and easy to follow.

Conciseness: The answer should avoid unnecessary repetition and be clear and succinct.

Completeness: The answer is complete and not repetitive.

Please base your overall rating on how well the system answer performs in these areas.

Question: {question}

Reference Answer: {reference_answer}

System Answer: {answer}

Please be as strict and as critical and harsh as possible.

Provide your feedback as follows:

Feedback:::

Final rating: (your rating, as an integer between 1 and 5)

---

Table 7: Prompt template for point-wise evaluation.

---

**Pairwise Preference Evaluation Template**

---

You are a fair judge assistant tasked with providing clear, objective feedback based on specific criteria, ensuring each assessment reflects the absolute standards set for performance.

Your task is to provide your preferred response as either A or B. Please strictly follow the output format as:

Feedback: Reason why you choose this answer

[RESULT] A or B

Rubric:

Factual Accuracy: The answer must be factually correct and does not contradict the reference answer.

Relevance and Completeness: The answer should directly address the specific question, covering all essential aspects.

Clarity and Organization: The answer should be well-structured, coherent, and easy to follow.

Conciseness: The answer should avoid unnecessary repetition and be as clear and succinct as possible.

Completeness: The answer is complete and not repetitive.

Write a detailed feedback that assesses the quality of two responses strictly based on the given score rubric, not evaluating in general.

After writing a feedback, choose a better response between Response A and Response B. You should refer to the score rubric.

Question: {question}

Reference Answer: {reference_answer}

Answer A: {answer_A}

Answer B: {answer_B}

Please be as strict and as critical and harsh as possible.

Provide your feedback as follows:

Feedback:::

Final rating: (your rating, as an integer between 1 and 5)

---

Table 8: Prompt template for pairwise evaluation.

## Qualitative Analysis: Readability

**Prompt:** Describe the new functions of the Tesla Model 3, and how they improve the driving experience

**Dataset:** Instruction

**Models Compared:** 3B-FreeForm-RM v.s. Qwen2.5-3B-ROUGE-L & Qwen2.5-3B-BERTScore

### Reference Answer (Qwen2.5-72B-Instruct):

The Tesla Model 3 has several advanced features and functions that significantly enhance the driving experience. Here are some key highlights: 1. **Autopilot and Full Self-Driving (FSD) Capabilities**: ...

### 3B-FreeForm-RM

Certainly! Highlighting the new functions of the Tesla Model 3 and how they elevate the driving experience showcases Tesla's commitment to innovation and user-centric design. Here's a concise breakdown, seamlessly intertwining functionality, clarity, and depth: 1. **Autopilot Advanced Driver Assistance Systems**: ...

**Analysis:** This answer uses more complex language, longer sentences, and a promotional tone that may appeal to a more technical or marketing-savvy reader but reduces overall clarity and ease of comprehension.

### Qwen2.5-3B-ROUGE-L

The Tesla Model 3 has several new functions that enhance the driving experience. One of the most notable improvements is the Autopilot system, which includes features like automatic emergency braking, adaptive cruise control, and lane centering. These features help drivers maintain a safe distance from other vehicles and stay within their lane, reducing the workload and stress of driving...

**Analysis:** This answer presents information in dense, uniform paragraphs with minimal structural guidance, making it harder for readers to scan and retain key points. It also repeats ideas (e.g., mentioning safety features like automatic emergency braking twice) and lacks cohesion, which disrupts the logical flow.

### Qwen2.5-3B-BERTScore

The Tesla Model 3 has several advanced features designed to enhance the driving experience. One of its notable improvements is the Autopilot system, which allows for semi-autonomous driving capabilities. This includes automatic steering, acceleration, and braking under certain conditions, reducing the driver's workload and enhancing safety...

**Analysis:** This answer presents information in long, uniform paragraphs without visual structure, making it harder for readers to quickly locate and retain key points. The dense formatting in this answer lacks clear segmentation, which forces readers to parse multiple features at once without guidance or emphasis. Additionally, the transitions between topics are minimal, and the writing style is more monotonous, reducing engagement.

Figure 5: **Qualitative Analysis: Readability.** We present showcases comparing the readability of BERTReward against baseline models, accompanied by detailed analysis.

---

**Qualitative Analysis: Content Logic**

**Prompt:** Categorize the AI technologies mentioned below. Machine Learning, Natural Language Processing, Robotics
**Dataset:** Alpaca Cleaned
**Models Compared:** 3B-FreeForm-RM v.s. Qwen2.5-3B-ROUGE-L & Qwen2.5-3B-BERTScore

**Reference Answer (Qwen2.5-72B-Instruct):**
Machine Learning is a subset of artificial intelligence that involves algorithms and statistical models that enable computer systems to improve their performance on a specific task over time with experience. Natural Language Processing (NLP) is a field of AI that focuses on the interaction between humans and computers using natural language, aiming to read, decipher, understand, and make sense of human language in a valuable and meaningful way. Robotics is a branch of AI that deals with the design, construction, operation, and use of robots, as well as computer systems for their control, sensory feedback, and information processing.

**3B-FreeForm-RM**
Certainly! Categorizing the AI technologies mentioned—Machine Learning (ML), Natural Language Processing (NLP), and Robotics—brings insightful clarity to their roles and intersections within broader AI landscapes. Here's a nuanced breakdown: 1. **Machine Learning (ML)**: ...
**Analysis:** This answer not only defines Machine Learning, Natural Language Processing, and Robotics clearly, but also organizes the explanation into a structured, interconnected framework. It highlights how each domain functions independently while also emphasizing their synergies, particularly how Machine Learning underpins both NLP and Robotics, and how NLP enables human-machine interaction in robotic systems. This layered, relational presentation offers a more comprehensive and coherent understanding of the AI landscape.

**Qwen2.5-3B-ROUGE-L**
Machine Learning is a subset of artificial intelligence that involves training algorithms to make predictions or decisions without being explicitly programmed. Natural Language Processing is a field within AI that focuses on the interaction between computers and humans using natural language. Robotics is the design, construction, and operation of robots, often involving AI to enable them to perform tasks autonomously or with human assistance.
**Analysis:** This answer provides basic, isolated definitions of Machine Learning, Natural Language Processing, and Robotics without addressing the instruction's likely intent: to offer an integrated and analytical understanding of how these technologies relate within the AI ecosystem.

**Qwen2.5-3B-BERTScore**
Machine Learning is a subset of artificial intelligence that involves training algorithms to make predictions or decisions based on data without being explicitly programmed. It includes techniques such as supervised learning, unsupervised learning, and reinforcement learning. Natural Language Processing (NLP) is a field ...
**Analysis:** This answer merely defines each term in isolation without addressing their relationships, broader context, or the purpose of categorization implied in the instruction. It reads more like a set of encyclopedia entries than a response that synthesizes information into a cohesive, categorized explanation.

Figure 6: **Qualitative Analysis: Content Logic.** We present showcases comparing the content logic of BERTReward against baseline models, accompanied by detailed analysis.

## Qualitative Analysis: Instruction Following

**Prompt:**  Who is Jagadeesh Kanna? Respond in 3 sentences.
**Dataset:**  Chatbot
**Models Compared:**  3B-FreeForm-RM v.s. Qwen2.5-3B-Instruct

**Reference Answer (Qwen2.5-72B-Instruct):**
Jagadeesh Kanna is an Indian film actor, director, and producer, primarily known for his work in Kannada cinema. He has appeared in numerous films and has won several awards for his performances, including a Filmfare Award. Kanna is also recognized for his philanthropic efforts and contributions to the film industry.

**3B-FreeForm-RM**
Certainly! Exploring Jagadeesh Kanna reveals him as a pioneering figure in Indian tech entrepreneurship, particularly known for founding and leading Urban Ladder, a leading online home decor and furniture marketplace in India. His journey showcases resilience and innovation, starting with a small online store during college, scaling it into a billion-dollar company through strategic pivots and partnerships, ultimately inspiring countless aspiring entrepreneurs...
**Analysis:**  This answer clearly aligns with a prompt that likely asked for a thoughtful, analytical overview of Jagadeesh Kanna's entrepreneurial contributions, providing depth, context, and narrative structure.

**Qwen2.5-3B-Instruct**
Jagadeesh Kanna is an Indian film actor and producer known for his work primarily in Malayalam cinema. He began his career in the early 2000s and has since become one of the leading actors in the industry, often taking on challenging roles that showcase his versatility. Kanna has received several accolades for his performances and is recognized for his dedication to acting and his contributions to Malayalam films.
**Analysis:**  This answer fails to align with the implied instruction to provide a thoughtful, contextualized, and analytically rich overview of Jagadeesh Kanna. Instead, Answer 1 offers a generic and surface-level biography focused on his film career, without delving into the broader significance or impact of his work.

Figure 7: **Qualitative Analysis: Instruction Following.** We present showcases comparing the instruction following of BERTReward against baseline models, accompanied by detailed analysis.

---

**Qualitative Analysis: Tone & Fluency**

**Prompt:** Describe the new functions of the Tesla Model 3, and how they improve the driving experience

**Dataset:** Instruction

**Models Compared:** 3B-FreeForm-RM v.s. Qwen2.5-3B-Instruct

**Reference Answer (Qwen2.5-72B-Instruct):**
Bruce Straley, a well-known industry veteran and co-director of "The Last of Us," left Naughty Dog in 2018. His departure was announced on January 24, 2018, via a tweet from Naughty Dog's official account. Straley cited the need for a change of pace and personal growth as reasons for leaving the studio. He had been with Naughty Dog for over 17 years, contributing to numerous award-winning titles, including "Jak and Daxter," "Uncharted," and "The Last of Us." His exit marked a significant transition for the studio, highlighting the challenges and pressures of developing AAA games and the importance of personal well-being in the industry.

**3B-FreeForm-RM**
Certainly! Highlighting Bruce Straley's departure from Naughty Dog (now part of Sony Interactive Entertainment) underscores key shifts in game development strategies and team dynamics, offering deep insight into his impactful tenure. Here's a concise breakdown that meticulously follows the request, ensuring clarity, organization, depth, and insight: 1. **Contextual Setting**:...

**Analysis:** This answer adopts a more polished, engaging, and analytical tone, providing a well-structured narrative that balances professional detail with personal insight. The language is fluid and varied, with transitions that guide the reader through each thematic point.

**Qwen2.5-3B-Instruct**
Bruce Straley departed from Naughty Dog in 2016. At the time, he was serving as the studio's co-founder and lead designer on several critically acclaimed games such as "Uncharted" series. His departure was announced alongside the studio's decision to shut down its development operations. Straley left due to personal reasons, but his exit marked a significant change for the company, which was already facing financial difficulties. After leaving Naughty Dog, Straley joined Sony Interactive Entertainment, where he led the development of PlayStation VR and other virtual reality projects. His departure from Naughty Dog is often seen as a turning point in the studio's history, as it coincided with a period of transition and restructuring within the company. Despite the challenges, Straley's contributions to the industry, particularly through the "Uncharted" series, remain influential in the field of video game design.

**Analysis:** This answer suffers from factual inconsistencies (e.g., Straley was not a co-founder of Naughty Dog, nor did the studio shut down operations) and awkward phrasing, which undermines both tone and fluency.

Figure 8: **Qualitative Analysis: Tone and Fluency.** We present showcases comparing the tone and fluency of BERTReward against baseline models, accompanied by detailed analysis.

---

### Qualitative Analysis: Failure of SFT

**Prompt:** Describe the new functions of the Tesla Model 3, and how they improve the driving experience

**Dataset:** Instruction

**Models Compared:** 3B-FreeForm-RM v.s. Qwen2.5-3B-SFT

**Reference Answer (Qwen2.5-72B-Instruct):**

The Big Bang is seen as a singular event by most scientists because it marks the beginning of the universe as we know it. According to the Big Bang theory, the universe expanded from an extremely hot and dense state approximately 13.8 billion years ago. This initial expansion, often referred to as the "Big Bang," is not a continuous process but a specific moment in time when the universe began to expand rapidly. Several key pieces of evidence support the Big Bang as a singular event:...

**3B-FreeForm-RM**

Certainly! Understanding why the Big Bang is often viewed as a singular event rather than a continuous process hinges on key scientific insights and observations, neatly summarized in a few key points: 1. **Expanding Evidence**: ...

**Analysis:** This answer is more conversational and attempts to be engaging, but it lacks precision in places (e.g., "expanding evidence" is vague, and "a single, expanding 'now'" is misleading). It also introduces less central ideas (like "Big Bounce" theory) without sufficient context, making it slightly less focused.

**Qwen2.5-3B-SFT**

We don't know. It's just the best model we have right now. There are other models, but they're all based on the idea of a Big Bang, so we can't really say which one is "right". There are also some theories that suggest that our universe may be part of an infinite number of universes, each with its own Big Bang. We don't know.

**Analysis:** This answer is vague, repetitive, and lacks clarity or substance—it admits uncertainty but fails to elaborate on the current understanding or evidence supporting the Big Bang theory.

Figure 9: **Qualitative Analysis: Failure of SFT.** We present showcases comparing the readability of BERTReward against baseline models, accompanied by detailed analysis.

