# OpenReview forum: "Taming Large Language Models for Free-Form Generation Via Reinforcement Learning With Verifiable Rewards"
_ICLR.cc/2026/Conference — ICLR 2026 Conference Withdrawn Submission_

### Official Review · Reviewer_4SYi · 2025-10-23

**Soundness:** 2
**Presentation:** 3
**Contribution:** 1
**Rating:** 2
**Confidence:** 5

**Summary:**

For open-ended tasks, it is often challenging to determine which responses are effective or not. This report proposes Free-Form, a reward model designed to construct reward signals for GRPO training. The results demonstrate that using a neural reward model to capture the relationship between open-ended responses and human ratings is more effective than relying on popular metrics such as ROUGE, BERTScore, and similar approaches.

**Strengths:**

- The paper is well-written.

- It clearly presents how the open-ended reward model is trained, providing sufficient background and illustrative figures that make it easy for readers to follow.

- Evaluating non-verifiable generation is an important and meaningful problem.

**Weaknesses:**

- **Lack of related work.** The authors only compare one form of the GRM, namely the pairwise version (see https://arxiv.org/pdf/2504.02495
).

- **Lack of novelty.** Using a reward model to evaluate open-ended tasks is already a common practice. This paper largely repeats existing training procedures without introducing substantial innovation.

- **Insufficient contribution discussion.** One claimed contribution is the proposal of the Free-Form reward model. However, the paper should devote more effort to explaining how it is constructed, how robust it is across model families, and how it addresses issues such as reward hacking and similar pitfalls. These points are only briefly mentioned, even though they are crucial and widely recognized challenges.

- **Lack of evaluation details.** The paper does not explain how it mitigates result fluctuations during sampling. Reporting metrics such as mean, variance, and significance tests would make the results more convincing.

**Questions:**

- Figure 3: It is unclear what 3B-GRM-llama-3B and 3B-PrefBert refer to. Please ensure consistent terminology throughout the paper.

- I am also curious why the base model and the reward model come from different model families. Some clarification or justification for this design choice would be helpful.

---

### Official Review · Reviewer_SoYm · 2025-10-31

**Soundness:** 3
**Presentation:** 2
**Contribution:** 1
**Rating:** 2
**Confidence:** 4

**Summary:**

This paper explores using GRPO to fine-tune LLMs on non-verifiable freeform generations. First, they train a reward model using Feedback Collection and Preference Collection (two existing LLM-as-a-judge and reward modeling datasets), and use the resulting rewards to fine-tune the LLM. They show that after fine-tuning the LLMs using GRPO, the average performance on three datasets outperforms the SFT baseline.

**Strengths:**

- The paper explores using GRPO to fine-tune LLMs on long-form generation tasks whose rewards are non-verifiable. This bridges the research gap, where current GRPOs are mostly for verifiable tasks like math, coding, or exact instruction-following.
- The writing of the paper is generally good, and the paper is easy to follow.

**Weaknesses:**

- **Highly misleading title and overall presentation**: The title says "*Taming Large Language Models for Free-Form Generation Via Reinforcement Learning With Verifiable Rewards*". However, the reward used in this paper is not a verifiable reward signal. It is the reward provided by a reward model, which is a neural network (or other rule-based metrics). It seems to me that the paper is confusing RLVR with GRPO. Using GRPO as the RL algorithm does not make it RLVR.
- **Limited novelty and significance**: This paper is essentially reinforcement learning with human feedback (RLHF), where they train reward models from human preferences and then use the reward model to fine-tune the policy model. The only distinction with past RLHF works is that in this paper, they select GRPO as the RL algorithm, while papers in the past use PPO (e.g., InstructGPT) or DPO. Changing the RL algorithm to an existing one is not a novel contribution.
- **Lack of important baselines**: Considering that the paper is essentially RLHF, the core contribution is how GRPO outperforms other RL algorithms (or not). As a result, some immediate baselines emerge, including PPO and DPO to train the policy. None of these methods are compared.
- **Unclear role of the BERTReward**: BERTReward is also a reward model fine-tuned on a reward modeling dataset, and it is used to train a policy model using GRPO. However, the paper includes this in Section 3.2 (existing popular methods for scoring open-ended generation) and puts the proposed (?) method, FreeForm-RM, in Section 3.3. It is quite unclear why these two methods are separated into two sections. These models are LMs trained on reward modeling or LLM-as-a-Judge datasets, and they are both neural reward models that take in a reference and generation and then output a scalar reward.

**Questions:**

The subject of the first sentence in Section 2 should be RLHFm, not RLVR. DPO and PPO should be cited in this section.

---

### Official Review · Reviewer_dbSm · 2025-11-01

**Soundness:** 2
**Presentation:** 2
**Contribution:** 2
**Rating:** 2
**Confidence:** 4

**Summary:**

This paper proposes using FreeForm–RM as the reward model for GRPO, which achieves superior performance on open-ended generation benchmarks (ELI5, Alpaca, LongForm) compared to larger counterparts and outperforms models trained with SFT in preference quality, as confirmed by both LLM-as-a-judge and human evaluation.

**Strengths:**

1. The paper uses RL method to verify the effectiveness of the reward model.
2. The paper presents comprehensive evaluations on open-ended generation benchmarks, including LLM-as-a-judge and human evaluation, showing that FreeForm–RM for GRPO outperforms larger counterparts and SFT-trained models.
3. The paper conducts thorough case studies and qualitative analyses, showcasing models trained with different approaches, including SFT, GRPO with traditional metrics, and GRPO with FreeForm-LM.

**Weaknesses:**

1. The work lacks novelty as it only adds a sigmoid function to normalize the score within the range of [0,1], making it no different from common reward models [2] or those used in PPO [1]. In addition, it does not include comparisons with other reward models on reward benchmark [3].
2. The sum of the Bradley-Terry Win Rates cannot equal 100 even when rounding is considered. For the Bradley-Terry model, a more common practice is to compute a score [3]. Moreover, using point-wise scores with the Bradley-Terry model is typically for score aggregation to perform **system-level ranking**, rather than for **pairwise preference evaluation** described in Section 5.1.
3. The paper misunderstands RLVR by assuming that a reward bounded between 0 and 1 combined with GRPO qualifies as RLVR. In essence, the approach still belongs to RLHF or RLAIF. Unless the reward is verified, for example through rule-based checks or code tests, it should not be categorized as RLVR.
4. "As shown in Figure 1, traditional metrics fail to distinguish between clearly better and worse responses, assigning similar scores to both, while GRM rewards often correlate with length rather than content quality." Statistical evidence is needed to support this claim.


[1] Stiennon, Nisan, et al. "Learning to summarize with human feedback." Advances in neural information processing systems 33 (2020): 3008-3021.

[2] https://huggingface.co/spaces/allenai/reward-bench

[3] https://github.com/lmarena/arena-hard-auto

**Questions:**

1. The classification tasks mentioned in the introduction and the dialogue and summarization tasks [1] mentioned in the related work also lack verifiable rewards and belong to RLHF or RLAIF, and therefore should not be regarded as RLVR.
2. According to the description in Section 3.3, FreeForm-RM is trained using a point-wise dataset with an MSE loss, whereas Appendix B.3 states that FreeForm-RM is trained with a pair-wise dataset and a Bradley-Terry loss. Which one is correct?
3. Figure 1 provides no explanation for GRM-llama-3B, and the FreeForm-LM shown in Figure 1 does not appear to correlate with human ratings. It is necessary to provide a correlation metric with human ratings computed over a sufficient number of samples.
4. Figure 3 provides no explanation for 3B-PrefBERT and 3B-GRM-llama-3B.
5. What is the purpose of the Pairwise Preference Evaluation Template shown in Table 8?

---

### Official Review · Reviewer_Q4bK · 2025-11-05

**Soundness:** 2
**Presentation:** 2
**Contribution:** 2
**Rating:** 2
**Confidence:** 3

**Summary:**

The paper studies the problem of scoring free form LLM responses for open ended tasks, specifically for online RL algorithms like GRPO which rely on rewards for each response in a group of candidate responses for a prompt. Previous techniques like ROUGE and BERTScore measure similarity with reference answers while BERTReward trains a BERT model with a regression head to predict the score. This paper proposes extending the latter technique to instead train a LLM Llama-3-3B model along with a regression head to predict the score, called FreeForm-RM.
The authors conduct experiments evaluating these reward models for GRPO on a dataset which is formed by a train mixture of three different free-form and open-ended task datasets with separate and overall test sets. They train two base Qwen2.5 models (1.5B and 3B), and use GPT-4 as a judge. The experiments show that  models trained using BERTReward and FreeForm-RM perform significantly better than the previous techniques, while among the former two the difference is smaller with FreeForm-RM being a bit better on most tasks. Similar results are obtained with human evaluation instead of GPT-4 as a judge.

**Strengths:**

1. The paper proposes a technique which scores free-form responses by using a Llama model along with a regression head trained on a free form rating quality dataset.
2. The experiments directly evaluate the efficacy of the proposed and previous methods on the downstream GRPO task.

**Weaknesses:**

1. The proposed FreeForm-RM method seems like a direct extension of BERTReward by replacing a BERT model with a Llama model, and as such seems to not have much methodological novelty or novel insights.
2. The gains of FreeForm-RM over BERTReward would be expected, given that a larger Llama model is used instead of BERT.
3. Both BERTReward and FreeForm-RM are trained on Prometheus datasets and thus their improved performance over baselines may not be surprising.
4. The paper’s writing and presentation can be improved (see questions to Authors).

**Questions:**

1. On line 290, the authors refer to BERTReward as “our lightweight BERTReward” which is confusing, what is lightweight about it? Also, since this is listed under Sec 3.2 which has existing techniques, why is it called “our … BERTReward” ?
2. The graphs in Figure 3 are very small and difficult to see. Also, the labels of the graphs should be made consistent with Table 1.
3. The caption of Figure 3 mentions that 3B-FreeForm-RM by step 60 generates 1024 tokens .. but this is not evident from the train/response_length graphs.

---

### Note · Authors · 2025-11-12

I have read and agree with the venue's withdrawal policy on behalf of myself and my co-authors.